# Predictive Value of Clinical and Dual-Energy Computed Tomography Parameters for Hemorrhagic Transformation and Long-Term Outcomes Following Endovascular Thrombectomy

**DOI:** 10.3390/diagnostics14222598

**Published:** 2024-11-19

**Authors:** Shiu-Yuan Huang, Nien-Chen Liao, Jin-An Huang, Wen-Hsien Chen, Hung-Chieh Chen

**Affiliations:** 1Department of Medical Education, Taichung Veterans General Hospital, Taichung 407219, Taiwan; stevenhuang19951203@gmail.com; 2Department of Neurology, Neurological Institute, Taichung Veterans General Hospital, Taichung 407219, Taiwan; rufus0822@gmail.com (N.-C.L.); jahuang@vghtc.gov.tw (J.-A.H.); 3Institute of Clinical Medicine, National Yang Ming Chiao Tung University, Taipei 112304, Taiwan; 4Department of Health Business Administration, Hungkuang University, Taichung 433304, Taiwan; 5Department of Radiology, Taichung Veterans General Hospital, Taichung 407219, Taiwan; chenws.tw@gmail.com; 6Department of Post-Baccalaureate Medicine, College of Medicine, National Chung Hsing University, Taichung 402202, Taiwan; 7College of Medicine, National Yang Ming Chiao Tung University, Taipei 112304, Taiwan

**Keywords:** dual-energy computed tomography (DECT), endovascular thrombectomy (EVT), acute ischemic stroke (AIS), ASPECTS

## Abstract

Objective: This study’s objective was to explore whether certain parameters measurable by dual-energy computed tomography (DECT) performed 24 h after endovascular thrombectomy (EVT) can predict subsequent hemorrhagic transformation. Material and Methods: We retrospectively reviewed patients with acute ischemic stroke (AIS) managed with EVT who had follow-up DECT within 24 h post-EVT between January 2019 and December 2023. Clinical and image parameters were recorded for predictive factor analysis. The primary outcome was hemorrhagic transformation, which was determined by using follow-up computed tomography (CT) or magnetic resonance imaging (MRI). The secondary outcomes were in-hospital mortality and 3-month post-EVT favorable functional outcome, as defined by a modified ranking scale (mRS) score of ≤2. Results: A total of 152 patients were included in this study. Multivariable analysis showed that the VNC-ASPECT score (*p* = 0.002) and superior sagittal sinus density (*p* = 0.01) were significantly associated with hemorrhagic transformation. For in-hospital survival rate analysis, post-EVT NIHSS measured 24 h post-EVT was an effective predictor, with a cutoff value of 23 (≤23: 88% vs. >23: 52.1%; *p* < 0.001). For functional outcome analysis, age (*p* < 0.001), tPA prior to EVT (*p* = 0.017), NIHSS 24 h post-EVT (*p* = 0.001), and VNC-ASPECT score (*p* < 0.003) were associated with a favorable functional outcome 3 months after EVT. Conclusions: The VNC-ASPECT score was associated with both hemorrhagic transformation and a 3-month post-EVT favorable functional outcome, and could therefore be an useful predictor for the development of hemorrhagic transformation.

## 1. Introduction

Endovascular thrombectomy (EVT) is the standard treatment procedure used for selected patients with acute ischemic stroke (AIS) [1]. However, hemorrhagic transformation (HT) is a relevant complication that occurs after EVT, and clinical treatment adjustment is needed [2,3,4]. Meanwhile, blood–brain barrier disruption with consequent hemorrhage or contrast staining is a well-known post-EVT situation, which may be difficult to differentiate from HT during follow-up conventional computed tomography. Hence, dual-energy computed tomography (DECT) plays a pivotal role due to its ability to effectively differentiate contrast staining from hemorrhage with high specificity and accuracy [5,6].

Previous studies have shown a variety of DECT applications after EVT. For instance, Gariani, Joanna et al. found that DECT-based virtual non-contrast (VNC) reconstructed imaging was superior to NCCT for the identification of acute ischemia after EVT based on the ASPECT ischemic region, with DECT performed 24 h after EVT [7]. Bonatti et al.’s team performed DECT immediately after EVT and demonstrated that the presence of parenchymal hyperdensity with a maximum iodine concentration > 1.35 mg/mL may identify patients developing intracerebral hemorrhage (ICH) [5]. Byre et al.’s team performed DECT within 1 h after EVT, showing that the relative percentage of iodine concentration at DECT compared with the superior sagittal sinus was a reliable predictor of ICH development [8]. Li et al.’s team performed DECT within 1 h after EVT, showing that the volume of the hyperdense area (HDA), the median maximum iodine concentration, and the maximum CT value showed great predictive performance in identifying ICH [9]. Ahn et al. demonstrated that all patients in their study underwent DECT immediately post-thrombectomy, showing contrast volume that can serve as a useful predictor of delayed parenchymal hematoma, which was associated with worse functional outcome and mortality [10].

Meanwhile, for the duration between EVT and follow-up DECT, previous studies suggest that most hemorrhagic transformations occur within 24 h after EVT, and the shorter the duration between EVT and follow-up DECT, the larger the proportion of changing diagnosis. Liu, Keqin et al. found that DECT performed both immediately and 24 h after EVT led to changes in the diagnosis of hemorrhagic transformation in a considerable proportion of acute ischemic stroke patients who underwent EVT. The study found that there was a lower proportion of diagnosis changes and better consistency 24 h after EVT [11]. Almqvist et al. found that performing DECT within 36 h after EVT led to changes in the radiologic reports regarding post-treatment ICH in a considerable proportion of patients undergoing EVT. They found that there were fewer new instances of ICH detected during DECT within the 18–36 h group compared to the group assessed < 18 h [12]. Hence, DECT performed within 24 h after EVT may be an appropriate method and duration.

Based on the various results of predictive value applied on DECT, most were performed within 1 h post-thrombectomy, as mentioned above, and no consensus was reached. Given the heightened risk of hemorrhage within the first 24 h following endovascular therapy (EVT) for patients suffering from stroke, we hypothesized that conducting a DECT scan within this critical period can predict subsequent hemorrhagic transformation. We also wanted to determine the relationship between image and clinical parameters and clinical outcomes for AIS patients managed with EVT.

## 2. Materials and Methods

### 2.1. Study Population

This retrospective study was approved by the Taichung Veteran General Hospital institutional review board (CE23526C), and informed consent was waived. Patients from January 2019 to December 2023 were reviewed. The inclusion criteria included the following: (1) AIS patient who received EVT; (2) the onset time of the symptoms being less than 24 h prior to EVT management; (3) a National Institutes of Health Stroke Scale (NIHSS) score prior to EVT ≧ 6; (4) DECT performed within 24 h after EVT; and (5) follow-up CT or MRI after EVT. Patients without a 3-month follow-up modified ranking scale (mRS) were excluded.

### 2.2. Imaging and Endovascular Treatment Protocols

Brain non-contrast CT (NCCT), CT angiography (CTA), and CT perfusion (CTP) were performed prior to EVT. Then, EVT was performed on a biplane X-ray system by a certified interventional thrombectomy specialist. All patients received EVT with a sequential approach according to our institutional protocols. Recanalization after EVT was evaluated using the modified thrombolysis in cerebral infarction (mTICI) scale [13]. Immediate post-EVT imaging was performed using cone beam CT. Within 24 h after EVT, DECT was performed on a dual-source CT scanner (IQon Spectral CT, Philips Healthcare, Eindhoven, The Netherlands), which generated mixed energy images (simulated 120 kV), virtual non-contrast (VNC) images, and iodine overlay map (IOM) images using a 3-material decomposition algorithm with commercially available software. Follow-up MRI or CT was performed to assess the development of hemorrhagic transformation. The timing of follow-up imaging was determined based on clinical decisions and the patient’s condition. All patients in our study were positioned supine on the table with their arms along their bodies.

### 2.3. Image Analysis

The patients underwent sequential image examination according to our institutional protocols. All images, including preoperative NCCT/CTA/CTP and post-operative DECT/CT/MRI, were interpreted by board-certified neuroradiologists. The initial assessment of DECT was performed by a post-graduate trainee and a neuroradiologist with 16 years of experience at our institution. Both reviewers evaluated the images, and any discrepancies were resolved through consensus.

For qualitative image analysis, the presence of a parenchymal hyperdense area (HDA) (yes/no) was assessed on simulated 120 kV images, which formed a simulated NCCT (sNCCT) image; the presence of iodine extravasation (yes/no) was assessed on iodine overlay map (IOM) images; and the presence of hemorrhage (yes/no) was assessed on virtual non-contrast (VNC) images. The initial appearance of hemorrhage was assessed on VNC images according to the European Cooperative Acute Stroke Study III (ECASS III) classification and categorized into hemorrhagic infarction (HI), parenchymal hematoma (PH), intraventricular hemorrhage (IVH), and subarachnoid hemorrhage (SAH). The final diagnosis of hemorrhagic transformation (HT) was made based on follow-up CT or MRI, which was our primary outcome and is detailed in the outcome measures section. The DECT condition was labeled as sNCCT (+) if the simulated 120 kV images presented with hyperdensity; as IOM (+) if there was apparent iodine extravasation on the IOM images; and as VNC (+) if a hemorrhage was present on the VNC images.

For quantitative image analysis, the HDA volume (mm^3^) and the average attenuation value (Hounsfield unit, HU) at the site of apparent attenuation (where the maximum value was recorded) were measured on simulated 120 kV images by drawing across the HDA if it was labeled as sNCCT (+) during DECT assessment. The average iodine concentration (milligrams/milliliter, mg/mL) and density (HU) at the site of the densest contrast staining (assessed and recorded subjectively) was measured on IOM images by drawing a 0.3 cm^2^ region of interest (ROI) circle if it was labeled IOM (+) during DECT assessment. The caudate nucleus (CN) average iodine concentration (mg/mL) and average attenuation value (HU) at the ipsilateral side of the occlusion vessel were measured on IOM/simulated 120 kV images by drawing a 0.3 cm^2^ ROI. Patients with hemorrhage or contrast staining at the CN were labeled as missing data. Superior sagittal sinus (SSS) average iodine concentration (mg/mL) and average attenuation value (HU) were measured on IOM/simulated 120 kV images by drawing a 0.1 cm^2^ ROI. The hypoperfusion area on the VNC images was scored according to the Alberta Stroke Programme Early CT Score (ASPECTS), thus forming a VNC-ASPECTS [14]. Due to different measurements between anterior and posterior circulation on ASPECTS, posterior circulation patients were labeled as missing data in our VNC-ASPECTS analysis. Ischemic volume (mm^3^) was measured on the DWI/ADC series during follow-up MRI examinations.

### 2.4. Clinical Variables

Clinical data were retrieved from our institutional electronic medical records and imaging system. Pre-procedural patient data, including age, sex, comorbid conditions, medication use, thrombolytic administration, NIHSS score at emergency room (ER NIHSS), pre-EVT laboratory data, and the site of vessel occlusion at preoperative CTA, were collected. Peri-procedural patient data, including details on percutaneous trans-arterial angioplasty (PTA) or stent usage, mTICI score, and the condition of cone beam CT evaluation performed immediately after EVT, were recorded. Post-procedural patient data were collected, including NIHSS score 24 h after EVT, neurosurgery intervention, in-hospital mortality, in-hospital length, mRS, and mortality 90 days after EVT. The 3-month mRS scores were assessed either during follow-up clinic visits or through structured telephone interviews.

### 2.5. Outcome Measures

The primary outcome was the presence of HT, which was defined by image diagnosis according to conventional CT or MRI performed after EVT and DECT to confirm HT. The secondary outcome was clinical, including in-hospital mortality during EVT admission and a functional outcome at 90 days after EVT; the favorable functional outcome was defined as mRS ≤ 2.

### 2.6. Statistical Analysis

For continuous variables, data are presented as median, and interquartile range (IQR). For categorical variables, data are presented as percentages. Comparison between subgroups for continuous variables was performed using the Mann–Whitney U test due to the non-normal distribution of the data. For categorical comparison, the Chi-Square test or Fisher’s exact test was applied.

Univariable logistic regression analyses were performed to assess the predictive value of different parameters on the outcome. Factors predictive in univariable analysis (*p* < 0.05) were further assessed in multivariable logistic regression analyses. *p* values of <0.05 were considered statistically significant. Receiver operating characteristic (ROC) curves were calculated for continuous variables significantly associated with HT. Statistical analysis was performed using SPSS, Version 29 (IBM, Armonk, NY, USA).

## 3. Results

### 3.1. Patient Characteristics and Clinical and Image Profiles

From January 2019 to December 2023, a total of 363 patients underwent EVT for acute stroke at our hospital. We excluded 185 patients who did not have DECT within 24 h, 24 patients who did not undergo follow-up CT or MRI before discharge, and 2 patients who lacked 3-month follow-up mRS scores. Ultimately, 152 patients were included in our analysis (Figure 1). There were 87 males (57.24%) and 65 females (42.76%), with a median age of 71.00 years (IQR 62.3–79 years), a median NIHSS score of admission of 17.00 (IQR 14–22), and a median NIHSS observed 24 h after EVT of 13.00 (IQR 7–18). For comorbid conditions, 103 patients (67.76%) with hypertension, 49 patients (32.24%) with diabetes mellitus, 106 patients (69.74%) with dyslipidemia, 89 patients (58.55%) with atrial fibrillation, and 28 patients (18.42%) with previous coronary artery disease were included. A total of 61 patients (40.13%) had received intravenous thrombolysis before EVT (Table 1). During EVT treatment, 136 patients (89.47%) presented with anterior circulation vessel occlusion, 21 patients (13.82%) received PTA only, and 9 patients (5.92%) received PTA with a stent.

For the procedural image profile, 119 patients (78.29%) with an mTICI score of 2B/2C/3 and 41 patients (26.97%) with hyperdensity at cone beam CT were treated immediately after EVT. For post-procedural DECT, 80 patients (52.63%) with IOM (−) and VNC (−) presented as sNCCT (−); 14 patients (9.21%) with IOM (+) and VNC (−); 27 patients (17.76%) with IOM (−) and VNC (+); and 31 patients (20.39%) with IOM (+) and VNC (+) were studied. The median VNC-ASPECT score was 7; the median HDA volume at sNCCT (+) was 2895 mm^3^; the median HDA density at sNCCT (+) was 46HU; the median IOM (+) area iodine concentration was 0.85 mg/mL; the median IOM (+) area density was 48.2 HU; the median caudate nucleus iodine concentration was 0.26 mg/mL; the median caudate nucleus density was 35.3 HU; the median superior sagittal sinus iodine concentration was 0.32 mg/mL; and the median superior sagittal sinus density was 43.25 HU. For post-procedural MRI, the median ischemic volume was 15,390 mm^3^ (Table 2), and the median time between EVT and MRI was 156 h.

There were two distinct time points for hemorrhage diagnosis in our study. The first time point was the initial hemorrhage assessed on VNC images, which were performed by DECT within 24 h after EVT. This initial assessment was categorized based on the European Cooperative Acute Stroke Study III (ECASS III) classification. The second time point was the final diagnosis of hemorrhagic transformation (HT), which was also our primary outcome and was assessed via follow-up CT or MRI based on clinical status and decisions. Initially, 58 patients were diagnosed with HT by VNC. By the time of discharge, a total of 94 patients (61.84%) were diagnosed with HT (including the initial 58). Among these, 55 patients were assessed by MRI, 12 were diagnosed through follow-up CT, and an additional 27 patients had findings consistent with HT on both MRI and follow-up CT. For secondary outcome measurements, 13 patients (8.55%) passed away during their in-hospital stay following EVT. Twelve patients (7.89%) had further surgical intervention, and forty-five patients (32.37%) had favorable functional outcomes at 3 months.

### 3.2. Clinical and Image Parameters: Hemorrhagic Transformation Development Analysis

To identify factors correlated with HT in our patient cohort, we compared clinical and imaging characteristics between the HT and non-HT groups. For clinical parameter analysis, PT (*p* = 0.007), NIHSS 24 h after EVT (*p* = 0.007), and surgical intervention after EVT (*p* = 0.004) were significantly different between the HT group and the group without HT. There was no difference between age, gender, atrial fibrillation, coronary artery disease, previous use of antiplatelets, tPA before EVT, NIHSS at admission, and post-EVT 3-month favorable mRS (Table 1).

For image parameter analysis, a pre-procedural CTP of T max > 6 s (*p* = 0.046), a post-procedural DECT VNC-ASPECT score (*p* < 0.001), the density of the caudate nucleus (*p* = 0.044), the density of the superior sagittal sinus (*p* = 0.003), and the post-procedural ischemic volume MRI were statistically significant between the HT group and the group without HT (Table 2).

### 3.3. Predictive Factors of Hemorrhagic Transformation Development: Univariate and Multivariate Analysis

For univariate regression analysis, NIHSS 24 h after EVT (*p* = 0.012), VNC-ASPECT score (*p* < 0.001), and density of superior sagittal sinus (*p* < 0.001) were associated with HT. For multivariate regression analysis, the VNC-ASPECT score (*p* = 0.002) and the density of the superior sagittal sinus (*p* = 0.01) were associated with HT (Table 3).

We analyzed a range of comorbidities to assess their predictive value for hemorrhagic transformation, including prior anticoagulant therapy, diabetes mellitus (DM), atrial fibrillation (AF), prior anti-coagulation use, and previous rTPA administration. As these parameters did not show significant differences, we presented only the main results in the manuscript for clarity. Detailed findings from the simple model logistic regression analysis of these comorbidities can be found in Appendix A.

We further performed a ROC curve for VNC-ASPECTS and the density of the superior sagittal sinus. For the VNC-ASPECT score, a cutoff value of 7 for identifying HT showed an area under the curve of 0.751, with 73% sensitivity (95% CI, 62.9%–81.9%) and 66% specificity (95% CI, 50.7%–79.1%) and a likelihood ratio of 2.15. For the density of the superior sagittal sinus, a cutoff value of 43.4 HU showed an area under the curve of 0.647, with 65% sensitivity (95% CI, 53.6%–74.8%) and 67% specificity (95% CI, 53.3%–79.3%) and a likelihood ratio of 1.98.

### 3.4. In-Hospital Mortality Analysis

For in-hospital mortality analysis, survival was statistically significantly associated with NIHSS 24 h after EVT (*p* = 0.003), triglyceride (*p* = 0.004), and HDL (*p* = 0.002) (Table 4). We further performed ROC to find the predictive cutoff value. A cutoff value of 23 at NIHSS 24 h after EVT showed an area under the curve of 0.748, with 46% sensitivity (95% CI, 19.2%–74.9%) and 94% specificity (95% CI, 88.7%–97.4%) and a likelihood ratio of 7.85. Further survival rate analysis showed a statistically significant difference with a 24 h NIHSS post-EVT cutoff value of 23 (≤23: 88% vs. >23: 52.1%, *p* < 0.001) (Figure 2).

We evaluated additional parameters related to in-hospital mortality, including the presence of HI, baseline NIHSS, and mTICI grade. Although none of these variables showed significant differences, we focused on presenting the most relevant results in the main text for clarity. Further details on the non-parametric analysis of these parameters are available in Appendix A.

### 3.5. Predictive Factors of a 3-Month Favorable Functional Outcome: Univariate and Multivariate Analysis

For univariate regression analysis, age (*p* < 0.001), tPA prior to EVT (*p* = 0.006), NIHSS at admission (*p* = 0.008), NIHSS 24 h post-EVT (*p* < 0.001), and the VNC-ASPECTS (*p* < 0.001) were associated with favorable functional outcomes at 3 months after EVT. For multivariate regression analysis, age (*p* < 0.001), tPA prior to EVT (*p* = 0.017), NIHSS 24 h post-EVT (*p* = 0.001), and VNC-ASPECTS (*p* < 0.003) were associated with favorable functional outcomes at 3 months after EVT (Table 5). For the VNC-ASPECT score, a cutoff value of 7 for predicting favorable functional outcome by ROC showed an area under the curve of 0.729, with 67% sensitivity (95% CI, 50.5%–80.4%) and 73% specificity (95% CI, 61.8%–81.8%) and a likelihood ratio of 2.43.

## 4. Discussion

The aim of our study was to explore whether certain parameters measurable by DECT performed 24 h after EVT can predict subsequent hemorrhagic transformation and clinical outcomes. We found that the VNC-ASPECT score, which was calculated based on the VNC images, could independently predict subsequential hemorrhagic transformation and a 3-month functional outcome post-EVT. We also found that post-EVT NIHSS measured after 24 h could predict the functional outcome, and values ≤23 were associated with a better survival rate.

### 4.1. VNC-ASPECT Score

Our study found that the VNC-ASPECT score could be an acceptable predictor for the development of HT and three-month favorable functional outcomes. The ASPECT score has been widely applied to AIS patients to evaluate the extent of the ischemic area and determine whether further treatment is needed [14]. The ASPECT score based on NCCT prior to EVT plays a role in predicting intraparenchymal hematoma detected immediately after EVT, which was associated with worse functional outcomes and mortality [16]. Moreover, Leker, Ronen R. et al. found that post-EVT ASPECTS ≥ 7 performed by NCCT correlates with good outcomes, with 86% sensitivity and 58% specificity [17]. These results were compatible with our study, with a VNC-ASPECT score cutoff value of 7 for identifying HT, showing an area under the curve of 0.751, with 73% sensitivity and 66% specificity. Previous studies found that VNC imaging could improve the qualitative and quantitative visualization of ischemic brain tissue in ischemic stroke patients after endovascular treatment. It could also improve the accuracy and diagnostic confidence in differentiating intracranial hemorrhage and contrast medium extravasation in acute stroke patients following intra-arterial revascularization [18,19,20]. However, only a few studies have focused on the application of DECT to ASPECT score evaluation. On VNC-ASPECTS, Van den Broek, Maarten et al. found that DECT angiography-derived VNC (DECTA-VNC) yielded similar ASPECT scores as NCCT; additionally, the inter-rater agreement was highest in the DECTA-VNC ASPECTS compared to NCCT [21]. Gariani, Joanna et al. found that VNC imaging was superior to NCCT for identifying acute ischemia after EVT based on the ASPECT ischemic region [7]. However, no study mentioned the predictive value of the VNC-ASPECT score; thus, ours may be the first study to report that VNC-ASPECTS based on DECT performed 24 h after EVT is an applicable method for the prediction of HT development and further management of AIS patients treated with EVT.

### 4.2. HDA Volume, HDA Iodine Concentration, and HDA Density

Our study identified that the volume of the hyperdense area on sNCCT and the maximum median density of the hyperdense area on sNCCT was not associated with hemorrhagic transformation. Our study also recognized that the density and iodine concentration based on contrast extravasation noted at IOM were not associated with either hemorrhagic transformation or functional outcomes. Previous studies have focused on the image parameters based on DECT, such as the volume or density of the hyperdense area on DECT; however, the inclusion criteria, the measurement of parameters such as the region of interest area (mm^2^), and the duration of DECT performed after EVT vary among different studies and could affect the results [5,8,9,10,22].

In our study, we have the highest number of patients, and DECT was performed within 24 h after EVT, in comparison to the articles mentioned above, where all DECT evaluations were performed less than 2 h after EVT. Our study showed that between the non-HT group and the HT group, there was no difference in HDA volume, the maximum median HDA density, the maximum iodine concentration on the IOM (+) area, and the maximum median IOM (+) area density. The amount of patients, the different inclusion criteria, and the DECT duration may be the causes of our different results, and further prospective studies might be needed. Moreover, our study found there was no difference between the non-HT and HT groups regarding the 3-month favorable functional outcome. However, there was a higher percentage in the non-HT group (*n*%, 37.74% vs. 29.07%, *p* = 0.289), possibly due to subsequential treatment adjustment after HT was noted.

### 4.3. Caudate Nucleus and Superior Sagittal Sinus Iodine Concentration and Density

In our study, there was a statistical difference in the median density of the caudate nucleus and the median density of the superior sagittal sinus between the non-HT group and HT group. Further univariate and multivariate analysis showed that the density of the superior sagittal sinus was associated with HT development. The lower the density of the superior sagittal sinus, the higher the risk of HT development. Combined with the results of the density and iodine concentration of the caudate nucleus and superior sagittal sinus, which showed lower values in the HT group, although not all of them were statistically significant, AIS or procedural-related blood–brain barrier damage may be the cause [23,24,25,26]. We hypothesize that, due to the association between HT and elevated intracranial pressure, there might be less blood flow into the brain and, subsequently, SSS, which causes differences in density between HT groups and non-HT groups; however, further pathophysiology studies are needed to prove this.

Byrne et al. found that the relative percentage of iodine concentration compared with the superior sagittal sinus was a predictor of intracerebral hemorrhage development [8]. The absolute concentration of iodine varies depending on the patient’s height, weight, cardiac output, renal function, contrast medium concentration, volume of contrast administered, and scan technique used [27]. Hence, normalizing the iodine concentration of brain parenchymal contrast staining using the superior sagittal sinus (SSS) would reduce the variability in iodine quantification related to systematic differences in post-EVT DECT [28,29]. Unfortunately, there were only 14 patients with IOM (+) VNC (−) in our DECT evaluation, and due to the small number of patients, we did not proceed with the analysis of the relative iodine concentration percentage between the IOM (+) area and the superior sagittal sinus.

### 4.4. Post-EVT NIHSS

Pre-EVT NIHSS scores are commonly used as predictors for intracerebral hemorrhage (ICH) after reperfusion therapies [30]; however, there was no association between pre-EVT NIHSS and HT development in our study. For post-EVT NIHSS performed 24 h after the procedure, it was not associated with HT development during multivariable analysis but instead associated with the 3-month favorable outcome. Jeong, Han-Gil et al. found that the post-EVT NIHSS score may be an appropriate baseline factor when evaluating an intervention after a hyperacute period [31]. Sajobi, Tolulope T. et al. found that the early trajectory of the NIHSS score within the first 48 h after EVT can help predict functional outcomes with high accuracy [32]. Lai, Yuzheng et al. found that the NIHSS score at 7 days post-EVT could serve as a marker predicting the outcomes for ischemic stroke patients [33]. The studies mentioned above were compatible with our study showing the potential value of post-EVT NIHSS score evaluation. Moreover, in our study, NIHSS at 24 h post-EVT with a cutoff value of 23 showed survival benefits post-EVT (≤23: 88% vs. >23: 52.1%, *p* < 0.001).

### 4.5. Limitations

Our study had several limitations. Firstly, due to its nature as a retrospective study conducted in a single institution, there was selection bias, with most of our patients being Taiwanese and thus lacking racial diversity and generalization. Secondly, although all patients received follow-up DECT within 24 h after EVT, the follow-up duration of MRI and CT was based on clinical decisions and status; hence, the duration varied among patients and might affect the diagnosis of hemorrhagic transformation. Furthermore, there is the possibility that patients who had asymptomatic hemorrhagic transformation remained undiagnosed. Due to the limitations mentioned above, further prospective studies are needed to validate our results, focusing on the precise duration of post-EVT imaging tools.

## 5. Conclusions

The VNC-ASPECT score was associated with both hemorrhagic transformation and a 3-month favorable functional outcome and could be an acceptable predictor for the development of hemorrhagic transformation. Moreover, NIHSS scores measured at 24 h post-EVT could be predictors of favorable functional outcomes. This information might be useful in further post-EVT treatment care and adjustment to minimize hemorrhagic complications.

## Figures and Tables

**Figure 1 diagnostics-14-02598-f001:**
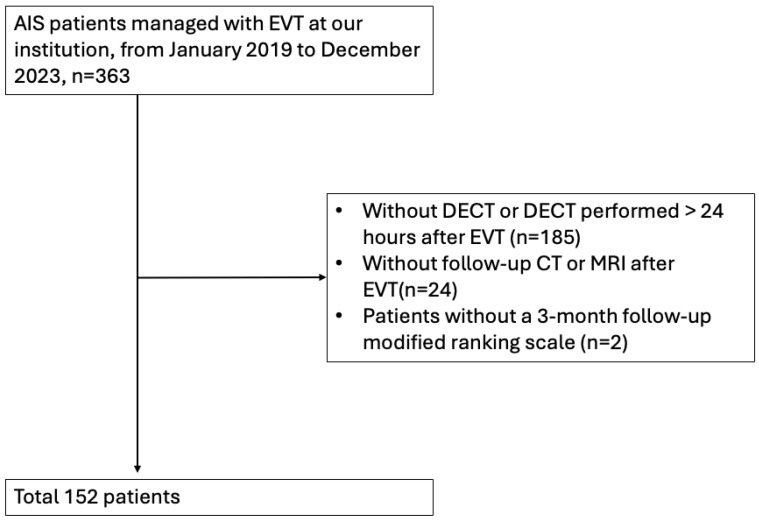
Patient selection flowchart for inclusion in the study.

**Figure 2 diagnostics-14-02598-f002:**
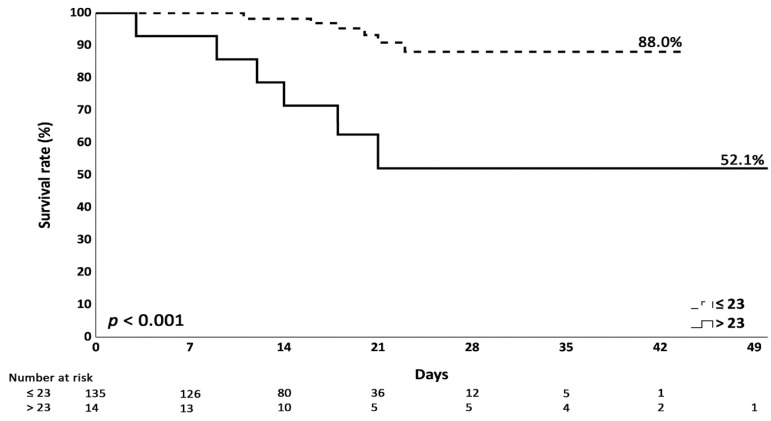
Kaplan–Meier survival analysis with a post-EVT 24 h cutoff value of 23.

**Table 1 diagnostics-14-02598-t001:** Clinical factors analysis of post-EVT AIS patients with or without HT.

	Total (*n* = 152)		Hemorrhagic Transformation				*p* Value
			No (*n* = 58)		Yes (*n* = 94)		
	Median	(IQR)	Median	(IQR)	Median	(IQR)	
Age	71.00	(62.3–79)	71.00	(62.8–80)	71.50	(61.3–78.3)	0.627
Gender, *n* (%)							0.344
Female	65	(42.76%)	22	(37.93%)	43	(45.74%)	
Male	87	(57.24%)	36	(62.07%)	51	(54.26%)	
Atrial fibrillation, *n* (%)	89	(58.55%)	32	(55.17%)	57	(60.64%)	0.506
Coronary artery disease, *n* (%)	28	(18.42%)	10	(17.24%)	18	(19.15%)	0.768
Previous usage of antiplatelet, *n* (%)	27	(17.76%)	7	(12.07%)	18	(19.15%)	0.253
Previous tPA, *n* (%)	61	(40.13%)	27	(46.55%)	34	(36.17%)	0.205
BMI (kg/m^2^)	23.96	(21.8–26.3)	24.61	(21.7–26.7)	23.42	(21.8–26.1)	0.434
PT	10.90	(10.5–11.4)	10.60	(10.3–11.2)	10.90	(10.6–11.5)	0.007 **
APTT	26.20	(24.5–28.3)	26.50	(24.7–28.5)	25.90	(24.3–28.3)	0.321
ER NIHSS	17.00	(14–22)	16.50	(12.8–21)	18.00	(14–22)	0.273
Post-EVT 24 h NIHSS	13.00	(7–18)	11.00	(5–16)	15.00	(8–19.5)	0.007 **
Circulation of vessel occluded *n* (%)							0.008 **
Anterior	136	(89.47%)	47	(81.03%)	89	(94.68%)	
Posterior	16	(10.53%)	11	(18.97%)	5	(5.32%)	
3-month favorable mRS, *n* (%)	45	(32.37%)	20	(37.74%)	25	(29.07%)	0.289
Surgical intervention, *n* (%)	12	(7.89%)	0	(0%)	12	(12.77%)	0.004 **

tPA tissue plasminogen activator. Mann–Whitney U test. Chi-Square test. Fisher’s exact test. ** *p* < 0.01.

**Table 2 diagnostics-14-02598-t002:** Image factors analysis of post-EVT AIS patients with or without HT.

	Total (*n* = 152)		Hemorrhagic Transformation				*p* Value
			No (*n* = 58)		Yes (*n* = 94)		
	Median	(IQR)	Median	(IQR)	Median	(IQR)	
**Pre-procedural CT**							
CBF < 30% of CTP	12	(0–32.1)	0	(0–24.5)	20	(0–37)	0.067
Tmax > 6 sec of CTP	84	(35.3–124.8)	72	(21.5–113)	89	(50–131)	0.046 *
**Procedural**							
mTICI, *n* (%)							0.126
0	18	(11.84%)	5	(8.62%)	13	(13.83%)	
1	5	(3.29%)	2	(3.45%)	3	(3.19%)	
2A	10	(6.58%)	1	(1.72%)	9	(9.57%)	
2B	29	(19.08%)	9	(15.52%)	20	(21.28%)	
2C	13	(8.55%)	8	(13.79%)	5	(5.32%)	
3	77	(50.66%)	33	(56.90%)	44	(46.81%)	
Cone beam CT, *n* (%)							0.008 **
Hemorrhage	14	(9.21%)	1	(1.72%)	13	(13.83%)	
Contrast Staining	27	(17.76%)	7	(12.07%)	20	(21.28%)	
Normal finding	111	(73.03%)	50	(86.21%)	61	(64.89%)	
**Post-procedural DECT**							
DECT condition, *n* (%)							<0.001 **
sNCCT (−)	80	(52.63%)	45	(77.59%)	35	(37.23%)	
IOM (+) VNC (−)	14	(9.21%)	8	(13.79%)	6	(6.38%)	
IOM (−) VNC (+)	27	(17.76%)	2	(3.45%)	25	(26.60%)	
IOM (+) VNC (+)	31	(20.39%)	3	(5.17%)	28	(29.79%)	
HT type of VNC, *n* (%)							<0.001 **
no HT	95	(62.50%)	53	(91.38%)	42	(44.68%)	
HI1	12	(7.89%)	5	(8.62%)	7	(7.45%)	
HI2	6	(3.95%)	0	(0%)	6	(6.38%)	
PH1	9	(5.92%)	0	(0%)	9	(9.57%)	
PH2	1	(0.66%)	0	(0%)	1	(1.06%)	
SAH	7	(4.61%)	0	(0%)	7	(7.45%)	
IVH	1	(0.66%)	0	(0%)	1	(1.06%)	
Combination of 2 HT types	16	(10.53%)	0	(0%)	16	(17.02%)	
Combination of 3 HT types	5	(3.29%)	0	(0%)	5	(5.32%)	
Hyperdense area on sNCCT (+) volume (mm^3^)	2895	(1086.3–7953.8)	1765	(510–19,422.5)	2925	(1220–7470)	0.392
Hyperdense area on sNCCT (+) density (HU)	46	(39–53)	46	(39.5–53)	46	(39–53)	0.895
IOM (+) area iodine concentration (mg/mL)	0.85	(0.7–1.4)	1.20	(0.9–1.5)	0.80	(0.6–1.1)	0.071
IOM (+) area density (HU)	48.20	(42.1–59.3)	50.90	(44.4–59.8)	47.10	(41.7–59.3)	0.491
Caudate nucleus iodine concentration (mg/mL)	0.26	(0.2–0.3)	0.27	(0.2–0.3)	0.26	(0.2–0.3)	0.699
Caudate nucleus density (HU)	35.30	(33.8–37.2)	36	(34.1–37.6)	35.05	(33.5–36.3)	0.044 *
Superior sagittal sinus iodine concentration (mg/mL)	0.32	(0.2–0.4)	0.34	(0.2–0.6)	0.30	(0.2–0.4)	0.078
Superior sagittal sinus density (HU)	43.25	(39.6–46.6)	44.60	(41.3–49.8)	42.20	(39.2–45.6)	0.003 **
VNC ASPECT score	7	(6–8)	8	(7–9)	7	(5–8)	<0.001 **
**Post-procedural MRI**							
Ischemic volume (mm^3^) by MRI	15,390	(4731–47,556)	9510	(2652–26,280)	20,730	(5862–56,430)	0.006 **

Mann–Whitney U test. Chi-Square test. Fisher’s exact test. * *p* < 0.05, ** *p* < 0.01.

**Table 3 diagnostics-14-02598-t003:** Predictive factors of HT.

	Simple Model			Multiple Model (*n* = 121)			
	OR	(95% CI)	*p* Value	OR	(95% CI)		*p* Value
Age	0.99	(0.96–1.02)	0.545				
Gender							
Female	1.00						
Male	0.72	(0.37–1.41)	0.345				
Previous tPA	0.65	(0.33–1.27)	0.206				
ER NIHSS	1.02	(0.97–1.08)	0.422				
Post-EVT 24 h NIHSS	1.06	(1.01–1.11)	0.012 *	1.02	(0.96–1.09)		0.475
Improvement of NIHSS	0.63	(0.30–1.35)	0.240				
CBF < 30% of CTP	1.01	(0.99–1.03)	0.243				
Tmax > 6 s of CTP	1.00	(1.00–1.01)	0.300				
VNC ASPECT score	0.52	(0.38–0.70)	<0.001 **	0.60	(0.43–0.83)		0.002 **
Hyperdense area on sNCCT (+) volume (mm^3^)	1.00	(1.00–1.00)	0.183				
IOM (+) area density (HU)	1.00	(0.97–1.04)	0.919				
Caudate nucleus density (HU)	0.89	(0.79–1.00)	0.052				
Superior sagittal sinus density (HU)	0.91	(0.85–0.97)	0.002 **	0.90	(0.83–0.98)		0.010 *

Logistic regression. * *p* < 0.05, ** *p* < 0.01.

**Table 4 diagnostics-14-02598-t004:** Analysis of in-hospital mortality.

	No (*n* = 139)		Yes (*n* = 13)		*p* Value
	Median	(IQR)	Median	(IQR)	
Post-EVT 24 h NIHSS	12	(7–18)	19	(13–33)	0.003 **
TOAST, *n* (%)					0.012 *
CE	94	(67.63%)	5	(38.46%)	
LAA	21	(15.11%)	1	(7.69%)	
Unknown or Other	24	(17.27%)	7	(53.85%)	
Triglyceride	72	(55–103.8)	103	(82–193.3)	0.004 **
HDL	47	(41–58)	37	(29–44)	0.002 **
PTA or Stent, *n* (%)					0.021 *
PTA only	21	(77.78%)	0	(0%)	
PTA with stent	6	(22.22%)	3	(100%)	

TOAST, Trial of ORG 10172 in Acute Stroke Treatment [15]. Mann–Whitney U test. Chi-Square test. Fisher’s exact test. * *p* < 0.05, ** *p* < 0.01

**Table 5 diagnostics-14-02598-t005:** Predictive factors of 3-month favorable mRS.

	Simple Model			Multiple Model (*n* = 123)			
	OR	(95% CI)	*p* Value	OR	(95% CI)		*p* Value
Age	0.94	(0.91–0.97)	<0.001 **	0.90	(0.85–0.95)		<0.001 **
Gender							
Female	1.00						
Male	1.51	(0.73–3.13)	0.264				
Previous tPA	2.80	(1.35–5.83)	0.006 **	3.67	(1.26–10.68)		0.017 *
ER NIHSS	0.92	(0.86–0.98)	0.008 **	1.01	(0.92–1.12)		0.779
Post-EVT 24 hr NIHSS	0.82	(0.76–0.89)	<0.001 **	0.84	(0.76–0.93)		0.001 **
VNC ASPECT score	1.86	(1.36–2.54)	<0.001 **	1.95	(1.25–3.04)		0.003 **
Hyperdense area on sNCCT (+) volume (mm^3^)	1.00	(1.00–1.00)	0.359				
IOM (+) area iodine concentration	0.62	(0.15–2.52)	0.504				
IOM (+) area density	0.98	(0.92–1.03)	0.408				
Caudate nucleus iodine concentration	0.28	(0.00–36.00)	0.606				
Caudate nucleus density	1.08	(0.96–1.21)	0.216				
Superior sagittal sinus iodine concentration	0.30	(0.05–1.80)	0.188				
Superior sagittal sinus density	0.99	(0.94–1.04)	0.637				
Ischemic volume (mm^3^) by MRI	1.00	(1.00–1.00)	0.111				

Logistic regression. * *p* < 0.05, ** *p* < 0.01.

## Data Availability

Data are available on request due to restrictions (e.g., privacy, legal or ethical reasons). The data presented in this study are available on request from the corresponding author due to IRB restriction.

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
