# Peer review of "Predictive Value of Clinical and Dual-Energy Computed Tomography Parameters for Hemorrhagic Transformation and Long-Term Outcomes Following Endovascular Thrombectomy"

_diagnostics, 2024, doi:10.3390/diagnostics14222598_

Round 1

Reviewer 1 Report

Comments and Suggestions for Authors

In this single-center retrospective study, the authors investigated whether certain imaging parameters measurable by dual-energy computer tomography (DECT) and generated 24 hours after endovascular thrombectomy (EVT) could predict subsequent hemorrhagic transformation (HT) in patients with acute ischemic stroke (AIS). Along with DECT imaging parameters, clinical parameters were also recorded from a total of 152 patients suffering from AIS and subsequently evaluated for predictive factor analysis. Key findings from the study were: (i) a multivariable analysis showed that the VNC-ASPECT score (calculated based on DECT VNC images) could independently predict the development of subsequent HT in this AIS population, (ii) an in-hospital survival rate analysis showed that the NIHSS score measured 24 hours post-EVT could predict the functional outcome with values ≤ 23 associated with better survival rates for these patients, and (iii) a functional outcome analysis showed that several parameters—i.e., patient age, tPA prior EVT, NIHSS score at 24 hours post-EVT, and the VNC-ASPECT score—are all associated with a favorable outcome at 3 months after the EVT procedure for AIS.    

Overall, I found the study to be not only well written, but also very informative and with clear conclusions supported by the included data, which I expect to greatly benefit the potential readers interested in this topic. The authors included a well-organized and comprehensive discussion section, which I also found very useful, as well as a paragraph that clearly explains the limitations of their study. My recommendation for the journal editors is to consider the manuscript for publication in its present form.  

Minor comment: For better clarity, my recommendation for the authors is to rephrase the sentence from lines 211-212 which currently reads “For secondary outcome measurements, 13 patients (8.55%) were mortal during EVT in-hospital admission”.  

Reviewer 2 Report

Comments and Suggestions for Authors

Comments:

This is a study evaluating the use of dual-energy computed tomography (DECT) to predict haemorrhagic transformation in patients with acute ischemic stroke after endovascular thrombectomy. In this single centre retrospective study, investigators have included 152 patients, and they report VNC -ASPECTS score and superior sagittal sinus density to be significantly associated with haemorrhagic transformation. The research question is not novel but is clinically important. The study design, analysis and their interpretations are appropriate. The manuscript is logically structured and easy to read. The authors need to address the following points:

1-       In the introduction part, while mentioning the rationale of the research question the authors mention that “…..conducting a DECT scan after this critical period can predict subsequent haemorrhagic transformation”. I think the authors intended to mention this ‘within’ the critical period of 24 hours. The inclusion criteria also mentions DECT within 24 hours of EVT. This should be corrected.

2-       While the Study title mentions use of DECT as predictive tool for haemorrhagic transformation, the primary outcomes are more of presence of HT and long-term outcomes of the cohort of post endovascular thrombectomy. The DECT parameters are more of a secondary analysis.

3-       Any sample size calculation if done for the study should be mentioned.

4-       Authors should mention the how the records were retrieved. Was it a electronic database or verification of physical records were done? The clinical outcomes e.g mRS , how were they assessed?

5-       A study flow chart should be provided mentioning all the patients who have undergone EVT during this time period and the exclusions and how the final subset were arrived at?

6-       When was the follow up MRI done? The time period post EVT should be mentioned.

7-       In the methods section the authors have mentioned that HI was diagnosed as per ECASS III classification. It is pertinent to mention that ECASS III was based in NCCT imaging. However, in this study authors mention 55 patients to be diagnosed as HI by MRI. The final diagnosis of HI was based on which imaging?

8-       The authors mention presence of HT by VNC as 57. But in results section the authors mention HI was diagnosed by CT in 12 patients. This should be explained.

9-       There were 89 patients in the cohort who had AF. Some of them would have been on prior anti-coagulations. Data regarding the same should be provided and analysed in the analysis to predict HI.

10-  The regression model has not considered important comorbidities e.g hypertension, diabetes, periprocedural factors e.g time to needle, periprocedural fluctuation in BP, anti-coagulation use in the analysis. These are important factors which can influence the occurrence of HI.

11-  The variables mentioned in the in-hospital mortality are very limited. Variable like Presence of HI, Baseline NIHSS, mTICI grade should be provided. Variables like TG and HDL do not follow any logical hypothesis.

Comments on the Quality of English Language

Minor editing of the English language is required.

Round 2

Reviewer 2 Report

Comments and Suggestions for Authors

The authors have addressed my previous queries adequately and necessary changes in the manuscript has been made.